# Impact of enhancing GP access to diagnostic imaging: A scoping review

Amy Phelan[1], John Broughan[2]*, Geoff McCombe[1], Claire Collins[3], Ronan Fawsitt[4,5], Mike O'Callaghan[6,7], Diarmuid Quinlan[6], Fintan Stanley[6], Walter Cullen[1]

1 School of Medicine, University College Dublin, Dublin, Ireland, 2 Clinical Research Centre, School of Medicine, University College Dublin, Dublin, Ireland, 3 Research, Policy and Information, Irish College of General Practitioners, Dublin, Ireland, 4 General Practice, Castle Gardens Medical Centre, Kilkenny, Ireland, 5 Primary Care Advisor, Ireland East Hospital Group, Dublin, Ireland, 6 Irish College of General Practitioners, ICGP, Dublin, Ireland, 7 School of Medicine, University of Limerick, Limerick, Ireland

* john.broughan@ucd.ie

**Data Availability Statement:** This paper is a scoping review, so all data were sourced from previously published material.

## Abstract

### Background

Direct access to diagnostic imaging in General Practice provides an avenue to reduce referrals to hospital-based specialities and emergency departments, and to ensure timely diagnosis. Enhanced GP access to radiology imaging could potentially reduce hospital referrals, hospital admissions, enhance patient care, and improve disease outcomes. This scoping review aims to demonstrate the value of direct access to diagnostic imaging in General Practice and how it has impacted on healthcare delivery and patient care.

### Methods

A search was conducted of 'PubMed', 'Cochrane Library', 'Embase' and 'Google Scholar' for papers published between 2012–2022 using Arksey and O'Malley's scoping review framework. The search process was guided by the PRISMA extension for Scoping Reviews checklist (PRISMA-ScR).

### Results

Twenty-three papers were included. The studies spanned numerous geographical locations (most commonly UK, Denmark, and Netherlands), encompassing several study designs (most commonly cohort studies, randomised controlled trials and observational studies), and a range of populations and sample sizes. Key outcomes reported included the level of access to imaging serves, the feasibility and cost effectiveness of direct access interventions, GP and patient satisfaction with direct access initiatives, and intervention related scan waiting times and referral process.

### Conclusion

Direct access to imaging for GPs can have many benefits for healthcare service delivery, patient care, and the wider healthcare ecosystem. GP focused direct access initiatives should therefore be considered as a desirable and viable health policy directive. Further

**Funding:** The authors received no funding for this work.

**Competing interests:** The authors have declared that no competing interests exist.

research is needed to more closely examine the impacts that access to imaging studies have on health system operations, especially those in General Practice. Research examining the impacts of access to multiple imaging modalities is also warranted.

## Introduction

Diagnostic imaging is vital in diagnosing and monitoring a wide spectrum of disease [1]. High demand on imaging services across the healthcare system poses intense pressure on limited diagnostic imaging resources. Research indicates that direct GP access to diagnostic imaging provides an avenue to timelier diagnosis, a consequent reduction in referrals to hospital-based specialists and emergency departments (ED) [2], and improved quality of patient care and disease outcomes [3, 4]. However, GPs' current ability to diagnose and treat public patients within General Practice is often limited by inadequate access to diagnostics that are frequently more readily available in hospital settings [2].

For example, in the Republic of Ireland (RoI), General Practitioners (GPs) are the first port of call for most medical problems and serve as the gateway through which patients access diagnostics and referral to hospital-based specialists. The RoI is unusual in Europe in that 42% of the population are eligible for free GP care [5] while the remainder of the population must pay for GP services. With limited options available for many public patients, GPs have previously been faced with patients requiring a certain imaging study (e.g., MRI or CT), yet such studies could only be organised by ED or other hospital-based specialists, which often caused delays. Therefore, enhanced access to diagnostics for public patients within General Practice, as has been facilitated via a number of Irish policy initiatives in recent years (see Table 1), has allowed GPs in Ireland to manage patients that would otherwise be referred to ED or outpatient clinics, thus, at least in theory, leading to reduced hospital referrals and/or admissions [6].

But limited resources and growing healthcare needs are a major concern, not only in Ireland, but also internationally. Indeed, workforce and workload challenges facing General Practice are well documented in the UK, the EU and further afield [12]. Furthermore, current evidence on the effects of improved access to diagnostic imaging in Primary Care is mixed. Some studies show that direct access is cost effective, timelier, and adept at identifying patient health problems [13, 14], while others state that direct access to imaging in primary care yields little to no benefit in terms of clinical or resource-based outcomes [15, 16]. The purpose of this scoping review is to provide clarity on this matter. The study will aim to do so by conducting

**Table 1. Direct access to diagnostic imaging in general practice schemes in Ireland.**

| | |
|---|---|
| *Community Based Diagnostics Initiative, 2007* [7] | Initiative to improve GP direct access to x-ray and ultrasound services. |
| *Sláintecare Reform 2017* [8] | Significant expansion of diagnostic services outside of the hospital setting as one of six critical changes to deliver efficient, effective, and integrated care. |
| *GP contract, 2019* [9] | Ensured an increase in direct access to imaging for all GPs as part of the phased I ntroduction of the structured management of chronic disease. |
| *Winter Planning Initiative 2020* [10] | Provided direct access by GP referral to x-ray, CT, MRI, dual energy X-ray absorptiometry (DEXA) and ultrasound to the full adult population of Ireland via private companies and hospitals. |
| *Enhanced Access to Diagnostics 2021* [11] | Permanent direct GP access to diagnostic imaging via outsourced services, available to all adult medial card and GP card holders. |

an inductive exploratory investigation of the current literature that demonstrates the value of, or lack thereof, enhanced direct access to diagnostic imaging in General Practice.

## Methods

A scoping review methodology was chosen to acquire a comprehensive overview of the literature regarding the value of enhanced access to diagnostic imaging in General Practice. Scoping review methods facilitate broad mapping of the literature, and they provide opportunity to identify key concepts and pertinent knowledge gaps. The scoping review framework used in this review consists of a six-stage process described by Arksey and O'Malley [17] with later recommendations by Levac et al [18]. A study protocol was not produced for this review.

### Stage 1: Identifying the research question

This scoping review aimed to determine the value of direct GP access to diagnostic imaging investigations. The following research question was formulated:

*'What does existing literature say about the value of enhanced access to diagnostic imaging in General Practice?'*

### Stage 2: Identifying relevant studies

A preliminary search of key databases was performed on the 10/06/2022. Online databases searched included PubMed/MEDLINE, Cochrane Library, Embase and Google Scholar. Multiple search terms were used to generate a reading list. For this, key words were identified, and medical subject heading (MeSH) terms were generated. The search terms were grouped, with results requiring mention of one search term in each group to be included (see below).

(('diagnostic imaging' [Title/Abstract]) **OR** ('x-ray' [Title/Abstract]) OR ('CT' [Title/Abstract]) OR ('CAT' [Title/Abstract]) OR ('MRI' [Title/Abstract]) OR ('DEXA' [Title/Abstract]) OR ('ultrasound' [Title/Abstract]) OR ('echocardiogram' [Title/Abstract]) OR ('radiology' [Title/Abstract]) AND ('general practice' [Title/Abstract])

Several additional relevant articles were identified by hand-searching references.

### Stage 3: Selecting studies

Titles and abstracts of identified studies were read by two reviewers (AP & JB) from the 13th–15th and 20th–21st June 2022 respectively, and those deemed relevant to the study were selected for full-text review. Full text reviewing was conducted by one reviewer (AP) from the 22nd–27th June 2022. The PRISMA Extension for Scoping Reviews (PRISMA ScR) flow diagram outlines the study selection process (Fig 1). Consistent with the scoping review methodology, inclusion criteria were broad to include a range of articles. Both peer-reviewed and grey literature were included. Literature was included irrespective of study design or methodology, resulting in various study types being included in the search. Once the initial search was performed, duplicates were removed, and studies were then included and excluded based on criteria described in Table 2.

### Stage 4: Charting the data

Once all relevant articles were identified, data were extrapolated and charted by one researcher to facilitate characterisation and thematic analysis of included studies. The following data was charted, as shown in Table 3:

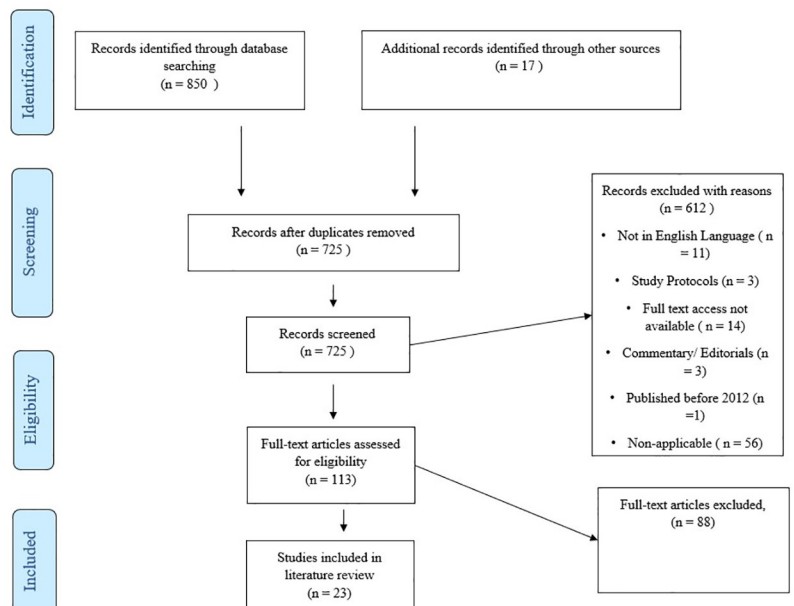

**Fig 1. PRISMA ScR flowchart.**

- Author, year of publication
- Journal/ publication
- Study title
- Study population
- Imaging modality
- Study location
- Study design
- Intervention
- Outcomes Measured
- Major findings

**Table 2. Study inclusion and exclusion criteria.**

| Inclusion Criteria | Exclusion Criteria |
|---|---|
| Published in English | Not available in English |
| Published between 2012–2022 | Published before 2012 |
| Focused on direct access referral to external diagnostic imaging services. | Did not focus on direct access referral to external diagnostic imaging services. |
| Peer reviewed and grey literature | Study protocols, commentaries & editorials |

## Stage 5: Collating, summarising, and reporting results

Data were collated, presented, and reported in the results section (see Table 3). Following this, major themes of the literature were identified using Braun and Clark's 'Thematic Analysis' approach [19]. The Thematic Analysis method facilitates a systematic and thus replicable approach to the coding, synthesis, and interpretation of qualitative data. Whilst Thematic Analysis is mostly used to analyse interview data, the method also provides a useful framework for analysing non-interview qualitative data such as that reported in scholarly articles. The method entails completion of six stages, these being (1) familiarisation with data, (2) initial code generation, the (3) searching for, (4) reviewing of, and (5) defining of themes, and (6) report writing. No assessment of methodological quality was performed. Efforts to establish the value of enhanced access to diagnostic imaging were guided by the Donabedian model for establishing quality of care, with the structure, process and outcomes of such enhanced access initiatives being examined [20].

## Stage 6: Consultation

In line with Levac et al.'s recommendations [18], experts in the field of General Practice were consulted to offer support with regards to the study's aims and conduct. Consulted personnel assisted with decision making around choices whether certain studies were to be included and excluded for review and interpretation of study findings. For this review, select GPs from the Irish College of General Practitioners and the University College Dublin / Ireland East Hospital Group GP Research Network were consulted on an ongoing basis throughout June to September 2022.

## Results

### Search results

Initial searches of the PubMed, Embase and Cochrane Library databases yielded 850 records published since 2012, with an additional 17 identified from hand searches. Following duplicate removal and reviewing of titles and abstracts, 113 were deemed relevant to the review and subject to full-text review. The search, identification and selection process are summarised in the PRISMA ScR diagram (Fig 1). Following this, 23 relevant papers were selected for final inclusion (Table 3).

### Study design

The 23 studies included in this review used various study types. There were six cohort studies [21–26], four randomised controlled trials [27–30], four observational studies [13, 31–33], three cross-sectional studies [34–36], three questionnaire studies [6, 37, 38], one retrospective analysis [39], one feasibility study [40], and one systematic review [3].

Seven studies were based in Denmark [21, 24–25, 28–30, 40], seven in the UK [3, 13, 23, 27, 36, 38, 39], six in the Netherlands [22, 26, 31–33, 35], two in Ireland [6, 37], and one in Italy [34].

The sample sizes of the studies ranged from 163 to 1,739,422 participants. Participants' ages ranged across studies. All included studies examined populations over 16 years old.

### Study population

Of the studies included, 17 examined populations with a specific diagnostic imaging modality. Five studies examined direct access to CT [25, 28–30, 36], four studies to MRI [22, 27, 31, 41], four studies to US [21, 24, 37, 40], three to echocardiogram (ECHO) [23, 26, 39], and one to

**Table 3. Description of studies included.**

| Author, Year | Journal/ Publication | Study Title | Imaging Modality | Study Population | Location | Study Design | Intervention | Outcomes Measured | Major Findings |
|---|---|---|---|---|---|---|---|---|---|
| Rutten et al, (2021) [32] | European Journal of General Practice | *Effects of access to radiology in out-of-hours primary care on patient satisfaction and length of stay* | - | Out-of-hours General Practices (n = 6) and patients presenting to-of-hours GP (n = 657) | The Netherlands | Multi-methods observational study (registration analysis and patient survey) | 3 models of direct access to diagnostic imaging services through General Practice: • unlimited direct access • limited direct access (restricted timeframe) • no direct access | • patient experience • length of stay | • direct access to diagnostic imaging services corresponds to a shorter length of stay for the patient • direct access to services results in higher patient satisfaction rates • patients felt taken seriously and had confidence in the expertise of those provides the service. |
| Smith et al. (2018) [3] | *British Journal of General Practice* | *Direct access cancer testing in primary care: a systematic review of use and clinical outcomes* | - | Systematic review of 60 papers | UK | Systematic Review | No intervention | • cancer conversion rate between direct access referral and referral through a hospital-based specialist • appropriateness of referrals -time interval from referral to testing • GP satisfaction • patient satisfaction | • patients satisfied by direct access services with the majority feeling it not necessary to see a hospital-based specialist prior to referral for testing • GPs felt direct access imaging both useful in diagnosis and cost-effective. |
| Appel et al, (2020) [21] | BMC Family Practice | *Direct-access to sonographic diagnosis of deep vein thrombosis in General Practice: a descriptive cohort study* | US | Patients presenting with suspected DVT in general practice (n = 449) | Denmark | Descriptive cohort study | Fast-track pathway for GPs to refer patients suspected of DVT directly to a same-day, whole leg compression US, without prior D-dimer test. Two strategies were available to GPs during referral: | • cost analysis pathway • referrals rate • findings on US and outcome of referral | • Direct-access to CUS for suspected DVT was achievable, had short time intervals and required fewer resources. • difference in DVT prevalence indicates that GPs distinguish between patients with low and high risk of DVT. |
| Rua et al, (2020) [13] | BMJ Open | *Management of chronic headache with referral from primary care to direct access to MRI compared with Neurology services: an observational prospective study in London* | MRI | Patients presenting with chronic headache in general practice (n = 249) | London, England | Observational prospective study | Chronic headache management pathway providing direct access to brain MRI through General Practice compared to an alternative pathway that facilitates referral from General Practice to the neurology department. | • 6-month healthcare costs associated with two existing clinical pathways • extension of the cost analysis up to 12 months • evaluation of access to care • patient satisfaction • headache burden and time off work associated with both clinical pathways | • at both 6- and 12-months post-recruitment, direct access to MRI for the management of chronic headache was associated with statistically significant mean cost savings for the NHS • participants in the neurology group reported higher levels of satisfaction due to increased time spent with clinical staff and feeling more informed about their condition • participants in both groups reported dissatisfaction with time between scan and the availability of the results. |

*(Continued)*

**Table 3.** (Continued)

| Author, Year | Journal/Publication | Study Title | Imaging Modality | Study Population | Location | Study Design | Intervention | Outcomes Measured | Major Findings |
|---|---|---|---|---|---|---|---|---|---|
| Berg et al, (2016) [31] | Family Practice | *Direct access to magnetic resonance imaging improved orthopaedic knee referrals in the Netherlands* | MRI | Patients presenting with knee pathology in general practice (n = 588) | The Netherlands | Observational study | Direct access to MRI of the knee by GP referral by GP at SHL-Groep in Etten-Leur, a diagnostic centre that provides support services to primary care in the region. | • findings on MRI<br>• management initiated 6 months post-MRI | • MRI for patients with knee complaints in the primary care setting significantly changed the pattern of GP referral to an orthopaedic surgeon.<br>• Direct GP access to MRI reduced the overall number of referrals to an orthopaedic surgeon in secondary care. |
| Chambers et al, (2014) [39] | British Journal of General Practice | *Detection of heart disease by open access echocardiography: a retrospective analysis of General Practice referrals* | ECHO | Review of open access ECHO of patients presenting with suspected heart disease in general practice (n = 2343) | Guy's and St Thomas' Hospital Trust, London, England | Retrospective analysis | Direct referral by GP for ECHO without direct involvement of a cardiologist | • findings on ECHO<br>• referral patterns | • open access echocardiography can detect heart abnormality that can alter patient management in 1/3 of cases |
| Zienius et al, (2019) [36] | BMC Family Practice | *Direct access CT for suspicion of brain tumour: an analysis of referral pathways in a population-based patient group* | CT | Review of direct access CT scans of patients presenting with suspected brain tumour in general practice (n = 2938) | Lothian region of Southeast Scotland | Population-based, cross-sectional study | Direct access to outpatient CT brain imaging via a single referral pathway | • findings on CT<br>• GP referral pattern based on presenting symptoms<br>• management of non-tumour findings by GP | • Kernick and NICE guidelines performed as predicted in indicating patients requiring further testing on suspicion of brain tumour and can be used in referral decisions.<br>• guidelines are insufficient in stratifying patients based on symptoms, with study suggesting that guidelines should be amended to better identify patients at risk of a brain tumour.<br>• direct access to such scans varies across the UK. |
| Guldbrandt et al, (2013) [30] | Danish Medical Journal | *Reduced specialist time with direct computed tomography for suspected lung cancer in primary care* | CT | Patients presenting with suspected lung cancer in general practice (n = 493) | Denmark | Randomised controlled study | Direct referral from GP through fast-track evaluation pathway for chest CT involving visit with chest specialist and chest CT. Half of patients recruited were randomly assigned to intervention while the other half went straight to chest CT before physician consultation. | • referral to CT<br>• conversion rate<br>• Chest specialist time per patient<br>• Staff acceptability | • direct access to CT reduced time with chest specialist<br>• direct access CT increased patient satisfaction<br>• most referrals by GPs were deemed appropriate by chest physicians |
| Ladegaard et al, (2021) [40] | Scandinavian Journal of Primary Health Care | *Direct access from General Practice to transvaginal ultrasound for early detection of ovarian cancer: a feasibility study* | Ultrasound | Patients presenting with suspected ovarian cancer in general practice (n = 479) | Denmark | Feasibility Study | Direct access to transvaginal US for GPs | • GP referral rate<br>• indications for referral<br>• management of test results<br>• findings on US | Providing GPs with direct access to transvaginal US was feasible<br>• 80% of the investigated women were referred back to the GP and managed within the primary care setting<br>• 14% referred on for further investigations and treatment |

*(Continued)*

**Table 3.** (Continued)

| Author, Year | Journal/ Publication | Study Title | Imaging Modality | Study Population | Location | Study Design | Intervention | Outcomes Measured | Major Findings |
|---|---|---|---|---|---|---|---|---|---|
| Moller et al, (2019) [25] | BMJ Open | *Diagnostic property of direct referral from general practitioners to contrast-enhanced thoracoabdominal CT in patients with serious but non-specific symptoms or signs of cancer: a retrospective cohort study on cancer prevalence after 12 months* | CT | Patients presenting with suspected cancer in general practice (n = 529) | Denmark | Retrospective Cohort Study | Patients directly referred by GP through the non-specific symptoms and signs cancer pathway for thoracoabdominal CT at Zealand University Hospital, Denmark. | • findings on CT<br>• GP referral patterns<br>• final diagnosis | • thoracoabdominal CT, as part of a GP-coordinated workup of NSSC, is an effective diagnostic tool<br>• referral rates corresponded to cancer incidence showing that GPs were appropriately referring patients |
| Fabich et al, (2016) [23] | British Journal of Cardiology | *'Quick-scan' cardiac ultrasound in a high-risk General Practice population* | ECHO | Patients presenting with suspected cardiac pathology in general practice (n = 163) | Lambeth and Southwark, England | Cohort Study | In one practice, patients referred by GP for 'quick scan' at one regular timeslot each week for 7 weeks. In second practice, patients referred by GP for 'quick scan' at two regular timeslots each week for 26 weeks | -finding on US | • direct access 'quick scans' can detect significant structural heart disease in patients within the primary care setting<br>• 'quick scan' may be a cost-effective triage method for patients with suspected heart failure |
| Nicholson et al, (2016) [38] | PloS One | *Variation in direct access to tests to investigate cancer: a survey of English general practitioners* | - | GPs (n = 533) | England | Survey | No intervention | • GP reported direct access to diagnostic imaging services | • almost all GPs had access to X-ray but access was much more varied for CT and MRI<br>• there was significant variation in access across regions of the NHS<br>• Apart from X-ray, very few GPs could access radiology within the timescales recommended by NICE. |
| Ladegaard et al, (2019) [24] | Acta Obstetricia et Gynecologica Scandinavica | *Ovarian cancer stage, variation in transvaginal ultrasound examination rates and the impact of an urgent referral pathway: A national ecological cohort study* | US | General Practices (n = 2769) and female patients presenting with suspected ovarian cancer in general practice (n = 1739422) | Denmark | Ecological Cohort Study | Direct access to transvaginal US through the standardised cancer patient pathway compared with access before the pathway was introduced in 2008. | • scan rates<br>• ovarian cancer incidence and stage | • US referrals increased with the implementation of the cancer patient pathway<br>• Prior the cancer patient pathway, women with the most access to transvaginal US were significantly more likely to be diagnosed with early-stage ovarian cancer compared with those less access<br>• direct access to US through the cancer patient pathway, eliminated this difference |
| de Schepper et al, (2016) [22] | Family Practice | *Prevalence of spinal pathology in patients presenting for lumbar MRI as referred from General Practice* | MRI | Patients presenting with spinal pathology in general practice (n = 683) | Rotterdam, The Netherlands | Cross-sectional, prospective, observational cohort study | Direct access to lumbar MRI from General Practice | • findings on MRI | • Almost all patients presenting for a lumbar MRI examination as referred by their GP had abnormal MRI findings |

*(Continued)*

**Table 3.** (Continued)

| Author, Year | Journal/ Publication | Study Title | Imaging Modality | Study Population | Location | Study Design | Intervention | Outcomes Measured | Major Findings |
|---|---|---|---|---|---|---|---|---|---|
| Hughes et al, (2015) [37] | Irish Medical Journal | *Open-access ultrasound referrals from General Practice* | US | GPs (n = 327) And review of open-access US scan referrals (n = 1090) | Ireland | | No intervention | • GP referral patterns • findings on scans • follow-up referral rate • final diagnosis | • Direct access US for general practitioners has been consistently shown to yield a similar rate of positive diagnostic outcomes to referrals generated from the hospital outpatient departments • Direct access to radiology results in an overall reduction in the number of referrals to hospital outpatient and emergency departments |
| Rutten et al, (2018) [33] | Family Practice | *Effects of access to radiology in out-of-hours primary care in the Netherlands: a prospective observational study* | - | General Practices (n = 6) Patients presenting to out-of-hours GP (n = 657) | The Netherlands | Prospective observational study | 3 models of direct access to diagnostic imaging services through General Practice: • unlimited direct access • limited direct access (restricted timeframe) • no direct access | • findings on scans • emergency department referral pattern | • direct access referral pathway results in 40% less referrals to the emergency department • patients with direct access referrals were more likely to follow-up treatment or visits |
| Guldbrandt et al, (2014) [29] | PloS One | *Implementing direct access to low-dose computed tomography in General Practice—method, adaption and outcome* | CT | General Practices (n = 119) with GPs (n = 266) | Denmark | Cohort study nested in a randomised study | Direct access to chest LDCT combined with a Continuing Medical Education (CME) meeting on lung cancer diagnosis. | • characteristics of patients referred • GP variation in use • amount of diagnostic work-up needed • cancer incidence | • 2/3 of GPs utilised the direct access pathway • GPs participating had a 61% higher referral rate • CME was associated with more than double positive predictive value. |
| Pertile et al. (2015) [34] | Cost Effectiveness and Resource Allocation | *Is chest X-ray screening for lung cancer in smokers cost-effective? Evidence from a population-based study in Italy* | X-ray | Patients presenting in general practice with increased risk of lung cancer (n = 1244) | Italy | Population-based study | Annual lung cancer screening by chest X-ray through General Practice for 4 years | • cost analysis of intervention | • direct access chest X-ray to detect for lung cancer is a cost-effective screening method in smokers • earlier detection of cancer results in improved 5-year survival rates with direct access X-ray |
| Schols et al, (2016) [35] | European Journal of General Practice | *Access to diagnostic tests during GP out-of-hours care: A cross-sectional study of all GP out-of-hours services in the Netherlands* | - | Out-of-hours GP practices (n = 117) | The Netherlands | Cross-sectional study | No intervention | • GP reported access to diagnostic imaging services | • direct access to diagnostic imaging is varied and limited during GP out-of-hours service • GP out-of-hours services adjacent to A&E departments do not offer wider access to diagnostic imaging, contrary to expectation |

*(Continued)*

**Table 3.** (Continued)

| Author, Year | Journal/ Publication | Study Title | Imaging Modality | Study Population | Location | Study Design | Intervention | Outcomes Measured | Major Findings |
|---|---|---|---|---|---|---|---|---|---|
| Van Gurp et al, (2013) [26] | Netherlands Heart Journal | *Benefits of an open access echocardiography service: a Dutch prospective cohort study* | ECHO | Patients presenting with suspected cardiac pathology in general practice (n = 155) and GPs (n = 138) | The Netherlands | Prospective Cohort Study | Open access echocardiography service through the SHL-Groep in Etten-Leur, a diagnostic centre which provides support services to primary care in the region. | • GP referral pattern<br>• findings on ECHO<br>• management initiated<br>• GP assessment of benefit of ECHO and cardiologist advice<br>• waiting time between positive scan and referral to cardiologist | • open access echocardiography may lead to significantly less referrals to the cardiologist<br>• more patients can be managed in primary care setting, with echocardiography aiding in decision making |
| Brealey et al, (2012). [27] | The British Journal of Radiology | *The effect of waiting times from general practitioner referral to MRI or orthopaedic consultation for the knee on patient-based outcomes* | MRI | General Practices (n = 163) and patients presenting with knee pathology in general practice (n = 553) | Urban, mixed and rural sites across northeast Scotland, north Wales and Yorkshire, UK | Secondary analysis of a randomised control trial | Direct access to MRI by GP to be performed within 12 weeks of referral. Educational seminars were also delivered to GPs about MRI, clinical diagnosis and conservative management of suspected internal derangement of the knee | • waiting time to MRI once referred | • direct access pathway to MRI resulted in a reduced waiting time of nearly 50% compared to the standard referral to orthopaedic specialist<br>• where a patient resides is a strong predicter in accessibility to diagnostic imaging services<br>• 86% of patient referred for MRI had a subsequent orthopaedic consultations, showing that GPs appropriately referred patients for imaging |
| O'Riordan et al, (2015) [6] | | *Access to diagnostics in primary care and the impact on a primary care led health service* | - | GPs (n = 292) | Ireland | Survey | No intervention | • GP reported direct access to diagnostic imaging services<br>• GP satisfaction with current services | • direct access and waiting times to MRI and CT differed significantly between the private and public system<br>• majority of participants believe that increased access to diagnostics would reduce emergency department referrals, also improving the quality of those referrals. This would subsequently reduce hospital admissions |
| Guldbrandt et al, (2015) [28] | Danish Medical Journal | *The effect of direct access to CT scan in early lung cancer detection: an unblinded, cluster-randomised trial* | CT | Patients presenting with suspected lung cancer in general practice (n = 331) | Denmark | Unblinded, cluster randomised trial | Direct access to low-dose CT in General Practice | • time between GP referral and scan<br>• time between first presentation to GP and definitive diagnosis<br>• stage of cancer at diagnosis | • direct access to low dose CT scans did not significantly influence stage or time of diagnosis of lung cancer compared to control arm of study.<br>• while the intervention may not have had the impact expected, it could serve as an alternative screening pathway to lung cancer |

X-ray [34]. The remaining six studies examined diagnostic imaging access in General Practice as a whole [3, 6, 32, 33, 35, 28]. Five studies focused on direct access to imaging in the context of managing non-specific and / or multiple different conditions [6, 32, 33, 35,37]. Four studies examined imaging for the management of lung cancer [28–30, 34], three focused on cancer in general terms [3, 25, 38], and two studied ovarian cancer [24–40]. Three studies concerned heart disease [23, 26, 39], and two focused on knee injury [27, 31]. Single studies examined direct access in relation to deep vein thrombosis [21], chronic headache [13], brain tumour [36], and spinal pathologies [22] respectively.

## Interventions studied

A key element across all studies was the implementation of access to external diagnostic imaging services through direct GP referral. The structure of the referral process varied between studies depending on the healthcare system involved and the nature of established access pathways. Most studies involved direct referral to imaging within an external radiology department, independent of hospital-based specialist involvement or additional testing [13, 21–34, 36, 39, 40]. In conjunction with enhanced access for GPs, two studies involved educational seminars for GPs on disease diagnosis and management [27, 29].

## Outcome measures

A range of outcomes were examined across included studies. One study assessed the feasibility of implementing a direct access pathway [40]. Another analysed utilisation of an established direct access pathway [29]. The cost effectiveness of direct access to diagnostic imaging was explored in four studies [13, 21, 23, 34]. Both GP and patient satisfaction with direct access referral pathways were examined in four studies [3, 13, 30, 32]. Time between GP referral to scan was assessed in three studies [21, 27, 32]. Of these, one study also examined time spent by patients at a scan through direct access referral [32]. Another study analysed times spent with hospital-based specialists due to a direct access CT pathway (30). Referral to EDs was examined in two studies [33, 37]. Four studies examined the appropriateness of such referrals [22, 25, 30, 37].

**Current direct access to diagnostic imaging services.** There was a consensus across studies that access to imaging services remained limited and varied widely across populations [6, 35, 36, 38]. In one study examining GPs and out-of-hours practices adjacent to hospital EDs, access to imaging was not shown to improve for out-of-hours services located adjacent to EDs, contrary to the study's expectations [35]. In Ireland, direct access to imaging differed considerably between public and private systems. This difference was accompanied by longer imaging study waiting times throughout the public system [6], regardless of whether the study was ordered by a GP or a hospital-based doctor. In the UK, those with direct access to imaging could not access such services within NICE recommended timescales [38].

**Cost-effectiveness.** Numerous studies found that direct access referral pathways for diagnostic imaging through General Practice are cost-effective, with more timely diagnosis and earlier treatment for patients being shown to further reduce overall costs within the healthcare system [13, 21, 34]. This was in part due to better use of hospital/radiology resources or reduced hospital admissions. No studies mentioned additional costs in terms of GP resources required to arrange and follow up on direct access imaging studies. One study determined direct ECHO access to be cost-effective with regards to triaging patients with suspected heart disease [23]. Another study showed that direct X-ray access in General Practice can serve as a cost-effective screening method for lung cancer in smokers [34].

**Patient satisfaction.**   It was found that patient satisfaction with diagnostic imaging services as part of diagnosis or management of their condition had increased with direct access through General Practice. Improved patient satisfaction was linked with reduced scan waiting times and referrals to hospital-based specialists facilitated by these interventions [3, 30, 32]. One study showed that alongside increased satisfaction, patients felt that they were taken seriously and had confidence in the expertise of those providing the services [32]. When compared to a control group with direct referral to a hospital-based specialist, patient satisfaction was lower than the control group, with the controls feeling more informed about their condition as a result of increased time spent with clinical staff [13]. In contrast, another study showed that most patients were satisfied with direct access to diagnostic imaging services and did not feel it necessary to see a hospital-based specialist prior to referral for testing [3].

**GP satisfaction.**   GP satisfaction was explored in one study, showing increased satisfaction because of direct access to imaging, which GPs felt was both useful in diagnosis and cost-effective. The additional workload and opportunity costs that the intervention placed on GP services was not mentioned when examining GP satisfaction [3].

**Feasibility and utilisation of intervention.**   One study explored the feasibility of a direct access transvaginal US pathway for early detection of ovarian cancer in General Practice [40]. The study revealed that such a pathway would be feasible. Eighty percent of patients were managed in Primary Care with the remaining 20% referred for further testing and visitation to a hospital-based specialist. Another study assessed the utilisation of a direct access CT pathway to diagnose lung cancer by GPs [29]. This study showed that two-thirds of GPs used the pathway once it was established. The reasons for lack of use by the remaining GPs were not examined.

**Appropriateness of referrals by GPs.**   By comparing referral patterns to subsequent findings on scans, several studies determined that GPs appropriately referred patients through direct access referral pathways [22, 25, 30, 37]. In one study, referrals from GPs and hospital outpatient departments yielded similar rates of positive diagnostic outcomes [37]. Further, GP direct access testing for symptoms that could indicate cancer has previously been criticised for increasing testing and decreasing diagnostic yield, but a systematic review examining this did not support these concerns [3]. This review reported that no significant difference was found in the cancer conversion rate between GP direct access and specialist testing pathways.

**Referral to hospital-based specialists and the emergency department.**   Both studies examining referrals to emergency departments showed a significant reduction in the number of patients referred to the ED following introduction of direct access referral pathways [33, 37]. Another study investigating referrals to hospital-based specialists highlighted that direct access to MRI in General Practice reduced the overall number of referrals to an orthopaedic surgeon in secondary care [31].

## Discussion

### Key findings

This scoping review's findings indicate that enhanced direct access to diagnostic imaging services within General Practice is a welcome, feasible, and with respect to health systems in their totality, a cost-effective measure that can often improve both system level and individual patient clinical outcomes (e.g., GP and patient satisfaction, scan waiting times, metrics illustrating referral processes). Both patients and clinicians have expressed satisfaction with direct access to imaging initiatives, particularly with regards to these interventions' positive impacts on waiting times for scans and referrals, diagnostic capacity, and clinical resource management. The level of GP and patient engagement with direct access interventions in the studies

examined was often high, indicating that diagnostic imaging interventions are well regarded by both patients and clinicians. The reviewed studies demonstrate that when supported with direct access, GP imaging referrals are generally appropriate, yielding high rates of positive diagnostic outcomes. Although, it should be acknowledged that appropriateness can be wide-ranging and multi-faceted in its meaning, and there is ongoing work to establish how best to appraise diagnostic imaging initiatives [42].

The findings also show that direct access to imaging can ensure more efficient use of technical and staff imaging resources, reduced hospital admissions, more timely diagnosis / earlier treatment for patients, and reduced overall costs within the health system. However, it is notable that much of the research examined focuses primarily on how direct access interventions impact on the health system at large and on hospital services. It is less clear how access to diagnostic imaging interventions impact on resources and costs specific to General Practice operations. It was also evident that most studies focused on the impacts of modality specific imaging with specific patient populations, that studies usually examined experimental direct access interventions rather than established frameworks, and that research comparing the accomplishments of direct access initiatives in public and private health systems, and across time, is lacking. Most studies included in this review documented direct access initiatives that were based in primarily state subsidised systems, and that were at a relatively early stage of implementation.

## Comparisons with existing literature

The World Health Organisation's overview of integrated care models describes community-based diagnostic imaging services as key to shifting the provision of care from acute to community settings [43]. Key Irish and international policy reports regarding direct imaging access initiatives for GPs share similar sentiments [7, 10, 12, 44]. This review's findings suggest that implementation of such initiatives is likely to have positive effects due to speedier diagnosis in the community and a more balanced sharing of responsibilities within the health system. This may in turn ease pressure on secondary care resources, and with respect to the Irish context in particular, it may contribute to reducing public outpatient waiting times. Further, in a 2015 survey, Irish GPs held that diagnostic imaging would improve patient care across a range of clinical scenarios [6]. This review's findings suggest that GPs accurately predicted the impact of such an initiative, especially with regards to the positive impacts on outpatient referrals and hospital admissions. Of course, not all researchers have communicated such positive views on enhanced GP access to imaging. For instance, Karel et al. (2015) contend that GP imaging referrals for knee and low back pain have ". . .little to no benefit" [15], while Sajid et al. (2020) claim that 'unfettered" GP access to imaging is conducive to mass wastage of service resource and therefore potentially harmful to patient health [16].

## Implications for future research and policy

This study's findings indicate that direct GP access to imaging can yield many benefits, particularly with regards to increased satisfaction levels amongst patients and doctors, reduced waiting times and referrals, feasibility of implementation, and cost-effectiveness within health systems at large. However, as mentioned, there is comparatively little research focusing on the impact that direct access initiatives have on General Practice workload and resources specifically, and future research regarding this issue is needed. Other important avenues for future research include examining the effects of direct access to imaging in general rather than with regards to condition specific patient populations. Furthermore, valuable insights may be gained by evaluating existing diagnostic imaging pathways rather than shorter-term

experimental pathways constructed solely for the purpose of research investigations. These findings indicate that health systems should continue to support diagnostic imaging pathways within Primary Care, for the benefit of patients, clinicians, and the overall health system. However, to make such initiatives sustainable, policymakers must also consider the opportunity cost of increased imaging responsibilities within General Practice, namely increased burden on already strained GP resources. Ageing populations and GP recruitment challenges have led to a steadily increasing GP workload in recent years, and displacement of other work by such initiatives may occur. As to whether this leads to unintended consequences, separate to effects of the imaging initiatives themselves, must be carefully assessed.

## Methodological considerations and limitations

The adoption of a scoping review methodology benefitted this study as the method permitted mapping of the literature concerning direct access to diagnostic imaging within Primary Care and General Practice settings, thus allowing us to provide a clear overview of a research topic that has not been widely investigated. Arksey and O'Malley's scoping review framework ensured that our research development, study selection, and data interpretation processes were conducted using an accepted and rigorous approach. There were some limitations to our review which should be considered. For instance, unlike a systematic review, a scoping review does not include an assessment of study quality as the focus is on covering the range of work that informs the topic rather than limiting the work to studies that meet pre-specified standards of scientific rigour. Further, only articles written in English and published in the last ten years on four electronic databases were considered for this review which may have resulted in the exclusion of relevant studies. For instance, all included studies were based in European countries and so understanding of direct GP access initiatives in non-European contexts remains a significant knowledge gap worth investigating in future research. Notwithstanding these limitations, the 23 studies included allowed us to gain a comprehensive overview of the current literature, identify research gaps, and inform future research on this topic.

## Conclusion

This study's findings suggest that direct access to diagnostic imaging services in General Practice may bring many advantages across the healthcare ecosystem. Going forward, health policy should seek to maximise the potential that direct access to imaging in communities can have for population health. Nonetheless, policymakers should also appreciate that continued research in this area, especially that which clearly delineates the various common imaging modalities and their outcomes, in addition to GP and patient perspectives, is required on an ongoing basis. Lastly, with respect to the Irish context, this study's findings indicate that reversion to previous pathways of imaging through hospital-based doctor referral will only lead to further delays in public healthcare, which is at odds with the stated aims of our health system's directives under Sláintecare.

## Acknowledgments

We would like to thank the Ireland East Hospital Group, the UCD School of Medicine, and the UCD College of Health and Agricultural Sciences. We would also like to thank staff at Affidea Diagnostics, especially Ms Muireann Feirtear, for their ongoing support of this research.

## Author Contributions

**Conceptualization:** Ronan Fawsitt, Walter Cullen.

**Formal analysis:** Amy Phelan, John Broughan.

**Funding acquisition:** Ronan Fawsitt, Walter Cullen.

**Investigation:** Amy Phelan.

**Methodology:** Amy Phelan, John Broughan, Geoff McCombe, Walter Cullen.

**Project administration:** John Broughan, Geoff McCombe, Walter Cullen.

**Resources:** Walter Cullen.

**Supervision:** John Broughan, Geoff McCombe, Walter Cullen.

**Visualization:** Amy Phelan.

**Writing – original draft:** Amy Phelan.

**Writing – review & editing:** Amy Phelan, John Broughan, Geoff McCombe, Claire Collins, Ronan Fawsitt, Mike O'Callaghan, Diarmuid Quinlan, Fintan Stanley, Walter Cullen.

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
