## [Decision Letter · Decision Letter 0]

31 Oct 2022

PONE-D-22-26921Impact of Enhancing GP Access to Diagnostic Imaging: A Scoping ReviewPLOS ONE

Dear Dr. Broughan,

Thank you for submitting your manuscript to PLOS ONE. After careful consideration, we feel that it has merit but does not fully meet PLOS ONE’s publication criteria as it currently stands. Therefore, we invite you to submit a revised version of the manuscript that addresses the points raised during the review process.

We look forward to receiving your revised manuscript.

Kind regards,

Tim Alex Lindskou

Academic Editor

PLOS ONE

Journal Requirements:

"The author(s) received no specific funding for this work. Having said this, the research group's general activities are supported via seed funding by the Ireland East Hospital Group and the College of Health and Agricultural Sciences / School of Medicine at University College Dublin, Ireland. These supports had no role in study design, data collection and analysis, decision to publish, or preparation of the manuscript."

"This study was funded by supports at the Ireland East Hospital Group, the UCD School of Medicine, and the UCD College of Health and Agricultural Sciences."

"The author(s) received no specific funding for this work. Having said this, the research group's general activities are supported via seed funding by the Ireland East Hospital Group and the College of Health and Agricultural Sciences / School of Medicine at University College Dublin, Ireland. These supports had no role in study design, data collection and analysis, decision to publish, or preparation of the manuscript."

5. Please include a caption for figure 1. 

7. We note that this manuscript is a systematic review or meta-analysis; our author guidelines therefore require that you use PRISMA guidance to help improve reporting quality of this type of study. Please upload copies of the completed PRISMA checklist as Supporting Information with a file name “PRISMA checklist”.

Reviewers' comments:

Reviewer's Responses to Questions

**Comments to the Author**

1. Is the manuscript technically sound, and do the data support the conclusions?

Reviewer #1: Yes

Reviewer #2: Yes

2. Has the statistical analysis been performed appropriately and rigorously? 

Reviewer #1: N/A

Reviewer #2: N/A

3. Have the authors made all data underlying the findings in their manuscript fully available?

Reviewer #1: No

Reviewer #2: Yes

4. Is the manuscript presented in an intelligible fashion and written in standard English?

Reviewer #1: Yes

Reviewer #2: Yes

5. Review Comments to the Author

Reviewer #1: Overall: This scoping review looked at the impact of Enhancing GP Access to Diagnostic Imaging. Authors introduce the topic well and demonstrate thorough knowledge of the area, as well as the methodological approach that followed the PRISMA extension for scoping reviews. My only question is about combining the focus on the tool’s effectiveness and implementation (access) of the tool in one review. They are related, but still separate phenomena and effectiveness (impact) is mostly an outcome of interest for systematic reviews because scoping reviews tend to be smaller and exploratory in nature to answer the definitive questions about effectiveness. I list some small suggestions for revisions of the methods and introduction section below.

- Limitations: would suggest including the absence of a published protocol of the scoping review.

Abstract:

- Please report the main outcome of interest in the abstract.

- While the abstract reports 23 studies being included in the review, the Figure 1 reports 25 studies; please clarify.

Introduction:

- I would suggest reducing the extent of the material presented from the current 10 paragraphs to 4-5 paragraphs maximum, perhaps excluding the content on Ireland given the international scope of the journal and its readership.

- p. 132: Purpose statement: there seems to be a discrepancy between how the aims of the review are described in the abstract: to gain a deeper understanding, vs. how they are presented in the purpose statement: “by evaluating current literature on the role and potential of enhanced direct access to diagnostic imaging in General Practice.”

Methods:

- Stage 1: I wonder whether a slightly re-phrased question would be more typical for or aligned with the scoping review methodology. Many scoping reviews ask: “What is the literature on [topic] like? Or what kind of studies have been published on [topic]?” rather than: has enhanced access to [tools] in [settings] improved healthcare service delivery and patient care.

- the scoping review framework is well chosen and described in the draft manuscript.

Results:

- n/a

Discussion:

- The discussion and conclusions stay within the confines of the findings of the study when they suggest potential usefulness of diagnostic imaging and recommendations for increasing access.

- P.17: Methodological considerations and limitations: the authors correctly identified the main restrictions on generalizability of their findings, including the English language criterion. One other limitation would be the limited number of the biomedical databases searched, as well as the 10-year time frame which makes the conclusions less “comprehensive (line 383)” and more “informative.” For instance, there seem to be no studies conducted in non-European countries, with the exception of the systematic review by Smith et al (2018) which reported that most studies were carried out in UK (MEDLINE, Embase, and the Cochrane Library were searched).

Reviewer #2: Thanks for the opportunity to review this manuscript. The authors detail a scoping review on the impact of access to diagnostic imaging in General Practice on healthcare service delivery and patient care. The review was well written and studied an important topic.

Below are a few suggestions to help improve the manuscript:

- It would be helpful to detail your inclusion criteria (even though it is broad) to help the reader understand exactly what kinds of articles you were interested in including (e.g. were descriptive studies included where no healthcare service delivery and/or patient care outcomes were measured?)

- Your measures could be categorized into Donabidian structure, process, outcome measures framework for understanding at a glance (via a visual) variety within study aims

- Were there any specific medical reasons or populations (e.g. cancer screening) that any of the studies described? This might be a helpful addition within the Study Population results section

- A comment on which countries the studies originated from would be helpful within the discussion – are the healthcare systems within these countries set up in the same way (or actively making such a transition)? For example, in Canada, GP referrals to diagnostic services has been the norm in many provinces and as a result, this might not be studied as often.

6. PLOS authors have the option to publish the peer review history of their article (what does this mean?). If published, this will include your full peer review and any attached files.

Reviewer #1: No

Reviewer #2: No

---

## [Author Response · Author response to Decision Letter 0]

16 Dec 2022

Journal Requirements

"The author(s) received no specific funding for this work. Having said this, the research group's general activities are supported via seed funding by the Ireland East Hospital Group and the College of Health and Agricultural Sciences / School of Medicine at University College Dublin, Ireland. These supports had no role in study design, data collection and analysis, decision to publish, or preparation of the manuscript."

Response

We have amended the cover letter to include a revised funding statement. The text reading “The authors received no funding for this work” is most applicable to this study, and the funding statement now reflects this. The ongoing support by the bodies mentioned (e.g., Ireland East Hospital Group, University College Dublin) for our research group is now mentioned in the acknowledgements section where it is more appropriate. 

"This study was funded by supports at the Ireland East Hospital Group, the UCD School of Medicine, and the UCD College of Health and Agricultural Sciences."

"The author(s) received no specific funding for this work. Having said this, the research group's general activities are supported via seed funding by the Ireland East Hospital Group and the College of Health and Agricultural Sciences / School of Medicine at University College Dublin, Ireland. These supports had no role in study design, data collection and analysis, decision to publish, or preparation of the manuscript."

Response

As noted above, we have amended the cover letter to include a revised funding statement and details of funding have been removed from the manuscript where applicable (e.g., acknowledgements section). The revised Acknowledgement section is as follows:

Page 17, line 375: “We would like to thank the Ireland East Hospital Group, the UCD School of Medicine, and the UCD College of Health and Agricultural Sciences. We would also like to thank staff at Affidea Diagnostics, especially Ms Muireann Feirtear, for their ongoing support of this research.”

Response

This study is a scoping review of the literature on our study topic. As such, data for this study is the contents of the published articles included in this review. The list of articles reviewed in this manuscript can be accessed via Table 3 and the reference list rather than a minimal dataset. Table 3 also outlines specific details of the data that was extracted from the included articles for the purpose of this study.

Response

As mentioned above, this study is a scoping review of the literature on our study topic. As such, data for this study is the contents of the published articles included in this review. The list of articles reviewed in this manuscript can be accessed via Table 3 and the reference list. Table 3 also outlines specific details of the data that was extracted from the included articles for the purpose of this study.

6. Please include a caption for figure 1. 

Response

A caption has been added to Figure 1

Response

Captions have been added for all supporting information files.

7. We note that this manuscript is a systematic review or meta-analysis; our author guidelines therefore require that you use PRISMA guidance to help improve reporting quality of this type of study. Please upload copies of the completed PRISMA checklist as Supporting Information with a file name “PRISMA checklist”.

Response

We have included a completed PRISMA extension for scoping reviews (PRISMA ScR) checklist with this submission. 

Response

We have reviewed the reference list to align with the revised version of the manuscript. 

The following reference which featured in the Introduction section of the original version was omitted because it did not align well with the changes made to this section.

Heavey P. The Irish healthcare system: A morality tale. Cambridge Quarterly of Healthcare Ethics. 2019;28(2):276-302. doi: 10.1017/S0963180119000100.

Reviewers' comments

1. Is the manuscript technically sound, and do the data support the conclusions?

Reviewer #1: Yes

Reviewer #2: Yes

2. Has the statistical analysis been performed appropriately and rigorously?

Reviewer #1: N/A

Reviewer #2: N/A

3. Have the authors made all data underlying the findings in their manuscript fully available?

Reviewer #1: No

Reviewer #2: Yes

Response

This study is a scoping review of the literature on our study topic. As such, data for this study is the contents of the published articles included in this review. The list of articles reviewed in this manuscript can be accessed via Table 3 and the reference list. Table 3 also outlines specific details of the data that was extracted from the included articles for the purpose of this study. 

4. Is the manuscript presented in an intelligible fashion and written in standard English?

Reviewer #1: Yes

Reviewer #2: Yes

Review Comments to the Author

1. Reviewer #1: 

 Overall: This scoping review looked at the impact of Enhancing GP Access to Diagnostic Imaging. Authors introduce the topic well and demonstrate thorough knowledge of the area, as well as the methodological approach that followed the PRISMA extension for scoping reviews. 

Response

Thank you for your feedback.

2. My only question is about combining the focus on the tool’s effectiveness and implementation (access) of the tool in one review. They are related, but still separate phenomena and effectiveness (impact) is mostly an outcome of interest for systematic reviews because scoping reviews tend to be smaller and exploratory in nature to answer the definitive questions about effectiveness. I list some small suggestions for revisions of the methods and introduction section below.

Response

We have amended the text accordingly throughout to communicate our study aims more clearly. We hope that this will alleviate concerns about the focus of the manuscript. The text has been amended to indicate that the scoping review has a broad exploratory aim in that it aims to synthesise literature that demonstrates the value of direct access to diagnostic imaging. Although we had expectations for the kind of content that we might find via our analysis, we did not define the parameters by which the value of direct access would be gauged a priori. Rather, we inductively extracted data from included studies that illustrated its value, and this data related to the content reported in the results section (e.g., feasibility, cost effectiveness, GP / patient satisfaction, waiting times, quality of referral processes, access)

Examples…

Abstract, Background, Page 2, Line 27-29: ‘This scoping review aims to demonstrate the value of direct access to diagnostic imaging in General Practice and how it has impacted on healthcare delivery and patient care.’

Introduction, Page 5, Line 93-94: ‘The purpose of this scoping review is to provide clarity on this matter. The study will aim to do so by conducting an inductive exploratory investigation of the current literature that demonstrates the value of, or lack thereof, enhanced direct access to diagnostic imaging in General Practice.’

Methods, Stage One: Identifying the research question, Page 5, Line 107-108: ‘This scoping review aimed to determine the value of direct GP access to diagnostic imaging investigations. The following research question was formulated: ‘What does existing literature say about the value of enhanced access to diagnostic imaging in General Practice?’

3. Limitations: would suggest including the absence of a published protocol of the scoping review.

Response

The absence of a published protocol has been noted in the manuscript (Methods, page 7, line 136). 

Methods, Page 5, Line 102-103: “A study protocol was not produced for this review.”

4. Abstract:

 Please report the main outcome of interest in the abstract.

Response

The results section of the abstract has been changed to include details on the main outcomes of interest. Several outcomes have been listed as concisely as possible there is not one main outcome of interest in the study.

Abstract, Page 2, Line 40-42: “Key outcomes reported included level of access to imaging services, the feasibility and cost effectiveness of direct access interventions, GP and patient satisfaction with direct access initiatives, and intervention related scan waiting times and referral processes.”

5. While the abstract reports 23 studies being included in the review, the Figure 1 reports 25 studies; please clarify.

Response: Thank you for highlighting this. The text in the figure was the error. There are 23 studies in the review as noted in the abstract and the figure has been amended to reflect this. 

6. Introduction:

 I would suggest reducing the extent of the material presented from the current 10 paragraphs to 4-5 paragraphs maximum, perhaps excluding the content on Ireland given the international scope of the journal and its readership.

Response

We have re-structured the introduction section to be briefer, less focused on the Irish context, and more attuned to justifying our study’s aim. No new evidence / points are presented. Making the changes involved deleting content, moving content from one paragraph to another, and adding a table (Table 1).

7. p. 132: Purpose statement: there seems to be a discrepancy between how the aims of the review are described in the abstract: to gain a deeper understanding, vs. how they are presented in the purpose statement: “by evaluating current literature on the role and potential of enhanced direct access to diagnostic imaging in General Practice.”

Response

As mentioned in response to Reviewer 1’s second comment, we have amended the text accordingly throughout to communicate our study aims more clearly. 

8. Methods:

 Stage 1: I wonder whether a slightly re-phrased question would be more typical for or aligned with the scoping review methodology. Many scoping reviews ask: “What is the literature on [topic] like? Or what kind of studies have been published on [topic]?” rather than: has enhanced access to [tools] in [settings] improved healthcare service delivery and patient care.

Response

Thank you for raising this point. We agree and have rephrased the review’s research question to something we believe is more appropriate.

Methods, Page 5, line 107-108: ‘What does existing literature say about the value of enhanced access to diagnostic imaging in General Practice?’

9. The scoping review framework is well chosen and described in the draft manuscript.

Response

Thank you

10. Results:

 - n/a

11. Discussion:

 The discussion and conclusions stay within the confines of the findings of the study when they suggest potential usefulness of diagnostic imaging and recommendations for increasing access.

Response

Thank you

12. P.17: Methodological considerations and limitations: the authors correctly identified the main restrictions on generalizability of their findings, including the English language criterion. One other limitation would be the limited number of the biomedical databases searched, as well as the 10-year time frame which makes the conclusions less “comprehensive (line 383)” and more “informative.” For instance, there seem to be no studies conducted in non-European countries, with the exception of the systematic review by Smith et al (2018) which reported that most studies were carried out in UK (MEDLINE, Embase, and the Cochrane Library were searched).

Response 

Details have been added to the Methodological considerations and limitations section regarding the limitations that you mentioned (i.e., limited number of the biomedical databases 

Discussion, Methodological considerations, page 17, line 357-361: ‘Further, only articles written in English and published in the last ten years on four electronic databases were considered for this review which may have resulted in the exclusion of relevant studies. For instance, all included studies were based in European countries and so understanding of direct GP access initiatives in non-European contexts remains a significant knowledge gap worth investigating in future research.’

Reviewer #2: 

Thanks for the opportunity to review this manuscript. The authors detail a scoping review on the impact of access to diagnostic imaging in General Practice on healthcare service delivery and patient care. The review was well written and studied an important topic.

Response

Thank you for your feedback.

Below are a few suggestions to help improve the manuscript:

1. It would be helpful to detail your inclusion criteria (even though it is broad) to help the reader understand exactly what kinds of articles you were interested in including (e.g., were descriptive studies included where no healthcare service delivery and/or patient care outcomes were measured?)

Response

A table has been added to the Methods section that clearly outlines the study’s inclusion and exclusion criteria. 

2. Your measures could be categorized into Donabidian structure, process, outcome measures framework for understanding at a glance (via a visual) variety within study aims

Response

This is an insightful comment and I have added text referring to how the study’s measures reflect the evaluation of healthcare from a Donabidian perspective in the Method section.

Methods, Page 8, Line 161-163:“Efforts to establish the value of enhanced access to diagnostic imaging were guided by the Donabedian model for establishing quality of care, with the structure, process and outcomes of such enhanced access initiatives being examined (23).”

3. Were there any specific medical reasons or populations (e.g. cancer screening) that any of the studies described? This might be a helpful addition within the Study Population results section

Response

We have added details regarding the specific medical reasons mentioned to the Study Population section.

Results, Study Population, Page 10, Line 195-201: “Five studies focused on direct access to imaging in the context of managing non-specific and / or multiple different conditions (6, 35, 36, 38, 40). Four studies examined imaging for the management of lung cancer (31-33, 37), three focused on cancer in general terms (3, 28, 41), and two studied ovarian cancer (27, 43). Three studies concerned heart disease (26, 29, 42), and two focused on knee injury (30, 34). Single studies examined direct access in relation to deep vein thrombosis (24), chronic headache (13), brain tumour (39), and spinal pathologies (25) respectively.”

4. A comment on which countries the studies originated from would be helpful within the discussion – are the healthcare systems within these countries set up in the same way (or actively making such a transition)? For example, in Canada, GP referrals to diagnostic services has been the norm in many provinces and as a result, this might not be studied as often.

Response

We have added details in the discussion section addressing the similarities and differences between the health systems of the included studies, and their implications for our interpretation of this study’s findings. 

Discussion, Key findings, Page 14, Line 306-310: ‘It was also evident that…research comparing the accomplishments of direct access initiatives in public and private health systems, and across time, is lacking. Most studies included in this review documented direct access initiatives that were based in primarily state subsidised systems, and that were at a relatively early stage of implementation.’

---

## [Editor Report · Decision Letter 1]

24 Jan 2023

Impact of Enhancing GP Access to Diagnostic Imaging: A Scoping Review

PONE-D-22-26921R1

Dear Dr. Broughan,

We’re pleased to inform you that your manuscript has been judged scientifically suitable for publication and will be formally accepted for publication once it meets all outstanding technical requirements.

Kind regards,

Tim Alex Lindskou

Academic Editor

PLOS ONE
---

## [Editor Report · Acceptance letter]

27 Feb 2023

PONE-D-22-26921R1 

Impact of enhancing GP access to diagnostic imaging: A scoping review 

Dear Dr. Broughan:

I'm pleased to inform you that your manuscript has been deemed suitable for publication in PLOS ONE. Congratulations! Your manuscript is now with our production department. 

Kind regards, 

on behalf of

Dr. Tim Alex Lindskou 

Academic Editor

PLOS ONE